# Chronological Contrastive Learning: Few-Shot Progression Assessment in Irreversible Diseases

**Clemens Watzenböck** [*1,2] iD          CLEMENS.WATZENBOECK@MEDUNIWIEN.AC.AT

**Daniel Aletaha**[3] iD **, Michaël Deman**[4]**, Thomas Deimel**[1,3]**, Jana Eder**[2,3] iD **, Ivana Janíčková**[1,2]**, Robert Janiczek**[4] iD **, Peter Mandl**[3,6] iD **, Philipp Seeböck**[1,2] iD **,
Gabriela Supp**[3]**, Paul Weiser**[1,2,5] iD **, Georg Langs**[*1,2] iD GEORG.LANGS@MEDUNIWIEN.AC.AT

[1] *Computational Imaging Research Lab, Department of Biomedical Imaging and Image-guided Therapy, Medical University of Vienna, Vienna, Austria,* [2] *Comprehensive Center for Artificial Intelligence in Medicine, Medical University of Vienna, Vienna, Austria,* [3] *Division of Rheumatology, Department of Medicine III, Medical University of Vienna, Vienna, Austria,* [4] *Johnson & Johnson,* [5] *Athinoula A. Martinos Center for Biomedical Imaging, Massachusetts General Hospital, Boston, Massachusetts, USA,* [6] *Ludwig Boltzmann Institute of Arthritis and Rehabilitation, Vienna, Austria*

**Editors:** Accepted for publication at MIDL 2026

## Abstract

Quantitative disease severity scoring in medical imaging is costly, time-consuming, and subject to inter-reader variability. At the same time, clinical archives contain far more longitudinal imaging data than expert-annotated severity scores. Existing self-supervised methods typically ignore this chronological structure. We introduce ChronoCon, a contrastive learning approach that replaces label-based ranking losses with rankings derived solely from the visitation order of a patient's longitudinal scans. Under the clinically plausible assumption of monotonic progression in irreversible diseases, the method learns disease-relevant representations without using any expert labels. This generalizes the idea of Rank-N-Contrast from label distances to temporal ordering. Evaluated on rheumatoid arthritis radiographs for severity assessment, the learned representations substantially improve label efficiency. In low-label settings, ChronoCon significantly outperforms a fully supervised baseline initialized from ImageNet weights. In a few-shot learning experiment, fine-tuning ChronoCon on expert scores from only five patients yields an intraclass correlation coefficient of 86% for severity score prediction. These results demonstrate the potential of chronological contrastive learning to exploit routinely available imaging metadata to reduce annotation requirements in the irreversible disease domain. Code is available at https://github.com/cirmuw/ChronoCon.

**Keywords:** Unsupervised Learning, Contrastive Learning, Few-Shot Learning, Representation Learning, Longitudinal Medical Imaging, Disease Progression, Rheumatoid Arthritis

## 1. Introduction

Time is of the essence in clinical settings. Time series – repeated scans of the same patient over multiple visits – capture essential information about disease evolution and treatment response. Although this information is routinely available in clinical archives, it is rarely used for representation learning. Most deep-learning approaches rely on large annotated datasets, yet expert scoring is expensive, time-consuming, and subject to inter-reader variability. In addition, discrete ordinal scores introduced to make expert assessment feasible and comparable capture only a coarse approximation of continuous disease severity. Often, they introduce quantization errors.

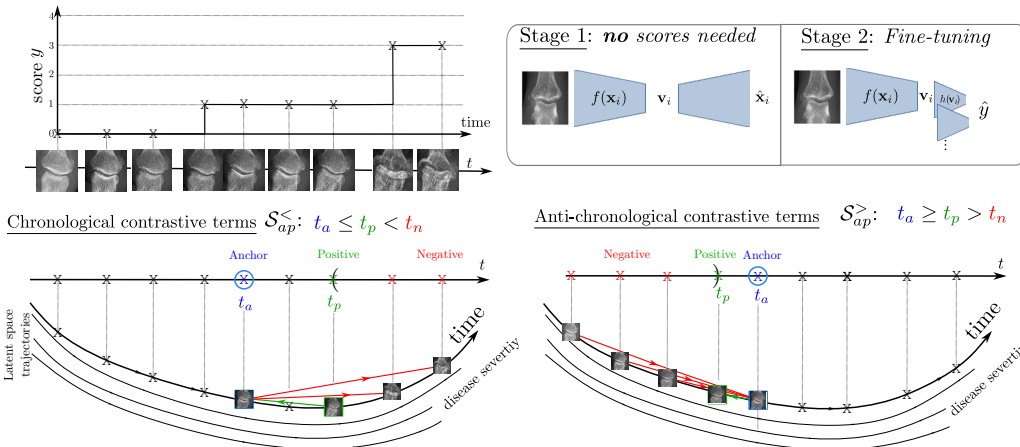

Figure 1: Chronological contrastive learning objective illustrated using a case of monotonically worsening joint-space narrowing (JSN) in a patient's interphalangeal (IP). *Bottom:* Anti-/chronological contrastive terms. The loss aligns disease trajectories in latent space, capturing severity automatically. *Top right:* Training stages. In stage 1, no labels beyond timestamps and patient+ROI IDs are required. In stage 2, the model is fine-tuned for score prediction.

We introduce *ChronoCon*, a chronological contrastive learning objective function that uses temporal examination order to train a model for mapping imaging data to quantitative severity scores. The idea is motivated by a simple example: consider a patient with an irreversible disease who is imaged at times $t_1 < t_2 < t_3$. In the latent-disease representation, the second scan should be at least as similar to the first scan as the third is to the first. Formally, for encoded features $\mathbf{v}_i$, we expect: $\text{sim}(\mathbf{v}_1, \mathbf{v}_2) \geq \text{sim}(\mathbf{v}_1, \mathbf{v}_3)$. A corresponding relation holds when comparing later visits to earlier ones. $\text{sim}(\mathbf{v}_2, \mathbf{v}_3) \geq \text{sim}(\mathbf{v}_1, \mathbf{v}_3)$. These ordering constraints, illustrated in Figure 1, allow the model to learn a progression-aware feature space without using any severity labels.

**Related work**   Recent work on image series and latent-space alignment incorporates temporal or pairwise information by jointly processing image pairs in both supervised (Kamran et al., 2025) and unsupervised settings (Bannur et al., 2023). (Kim and Sabuncu, 2023), also assumes monotonic progression, just as we do, but operates on image pairs using a learned classifier to predict temporal order. Likewise, (Chakravarty et al., 2024) enforce increasing risk scores via pairwise losses and parallel hyperplanes in latent space. While effective at capturing pairwise differences, these approaches do not leverage the full temporal trajectory available in longitudinal patient data.

(Holland et al., 2024) define positives as visits from the same patient within a predefined time window and negatives across patients. This requires known progression timescales, frequent acquisitions, and balanced disease states—assumptions that may not hold in many longitudinal settings such as rheumatoid arthritis progression.

In contrast to all these approaches, ChronoCon does not operate on image pairs or fixed time windows, but leverages the complete visit sequence to impose ordering directly in latent space without additional learnable components.

(Zeghlache et al., 2024) combine self-supervision with Neural ODEs to model continuous disease dynamics and naturally handle irregular sampling. This more general formulation assumes differentiable feature evolution and adds ODE training complexity, whereas Chrono-Con makes no continuity assumptions and is suited for trajectories with abrupt changes.

In supervised contrastive learning, several methods define positive and negative pairs based on label ordering (Gong et al., 2022; Zha et al., 2023). (Janíčková et al., 2025) used a triplet loss with time-dependent margins as hyperparameters, which makes the approach difficult to apply to nonlinear progressions in irregularly sampled time series. Conversely, while (Couronné et al., 2021) handles irregular sampling, the soft-rank loss does not enforce discriminability across more distant visits: it preserves ordering without explicitly pushing farther-apart time points away in latent space, similar to label-distribution smoothing or feature-distribution smoothing (Yang et al., 2021).

The closest work to ours is *Rank-N-Contrast* (RnC) (Zha et al., 2023). RnC defines the conditional probability that the positive $(p)$ is the correct match for the anchor $(a)$ among its negatives $n \in \mathcal{S}_{ap}^{\bullet}$ as

$$P(\mathbf{v}_p \mid \mathbf{v}_a, \mathcal{S}_{ap}^{\bullet}) = \frac{\exp\big[\mathrm{sim}(\mathbf{v}_a, \mathbf{v}_p)\big]}{\exp\big[\mathrm{sim}(\mathbf{v}_a, \mathbf{v}_p)\big] + \sum_{n \in \mathcal{S}_{ap}^{\bullet} \setminus \{p\}} \exp\big[\mathrm{sim}(\mathbf{v}_a, \mathbf{v}_n)\big]}. \tag{1}$$

The corresponding per-pair loss is $\ell_{ap}^{\bullet} = -\log P(\mathbf{v}_p \mid \mathbf{v}_a, \mathcal{S}_{ap}^{\bullet})$. Negatives are selected based on distances in label space $\mathcal{S}_{ab}^{\mathrm{RnC}} := \big\{ k \mid k \neq i, \; |y_a - y_n| \geq |y_a - y_p| \big\}$, a strategy well suited for fully supervised regression problems. RnC has since been applied to sentiment analysis (Weng et al., 2025), visual-concept explanation (Obadic et al., 2024), and extended to survival prediction (Saeed et al., 2024). However, neither RnC nor these generalizations[1] can be used *without labels* and thus cannot be applied directly to timestamps.

For a patient with visits at $t_1 = 0$, $t_2 = 1$, and $t_3 = 3$ years, RnC would imply that features between first and second visit are more similar than those between the second and third. In irreversible diseases, however, progression is nonlinear: long periods of stability may be followed by abrupt worsening. Consequently, absolute time intervals are not meaningful distances.

## 2. Methods

For each image $\mathbf{x}$, the corresponding relative time point $t$ within a patient's examination series is available. For some images, an additional ordinal expert-annotated score $y$ is provided. We use this information to encourage representations that capture disease progression. Let $\mathcal{D} = \{(\mathbf{x}_i, t_i, y_i, \mathtt{id}_i)\}_{i=1\ldots N}$ denote a dataset of $N$ examples with imaging, time, and scoring information, where $\mathtt{id}_i$ is the group identifier determining which samples may be contrasted against each other. We propose a two-stage learning procedure. In the *first stage*, only imaging data and metadata are required. We learn a mapping $f : \mathbb{R}^{l \times w} \to \mathbb{R}^d$, $\mathbf{x} \mapsto \mathbf{v}$ from

---

1. In (Saeed et al., 2024), time-to-event was also the prediction target, and repeated imaging for the same patient was not considered.

the image space to a latent representation space using ChronoCon. The goal of the *second stage* is to learn a scoring function, mapping latent representations to an estimate of the ordinal score, $h : \mathbb{R}^d \to \mathbb{R}$, $\mathbf{v} \mapsto \hat{y}$, trained using only an MSE loss.

**Chronological contrastive learning.** To apply contrastive learning to time-stamps, we introduce the sets of chronological negatives $\mathcal{S}_{ap}^<$ and anti-chronological negatives $\mathcal{S}_{ap}^>$ as

$$
\begin{aligned}
\mathcal{S}_{ap}^< &= \{n \mid n \neq a, \ \mathrm{id}_a = \mathrm{id}_p = \mathrm{id}_n, \ (t_a \leq t_p < t_n) \}, \\
\mathcal{S}_{ap}^> &= \{n \mid n \neq a, \ \mathrm{id}_a = \mathrm{id}_p = \mathrm{id}_n, \ (t_a \geq t_p > t_n) \}.
\end{aligned}
\tag{2}
$$

Trivial pairs without valid negatives are excluded from normalization, and we define *ChronoCon* loss as the balanced[2] sum of the forward and backward chronological contributions:

$$
\begin{aligned}
L^{\mathrm{ChronoCon}} &= \frac{1}{|\mathcal{P}_+^<|} \sum_{(a,p)\in\mathcal{P}_+^<} \ell_{ap}^< + \frac{1}{|\mathcal{P}_+^>|} \sum_{(a,p)\in\mathcal{P}_+^>} \ell_{ap}^>, \\
\mathcal{P}_+^< &= \{(a,p) \mid \mathrm{id}_a = \mathrm{id}_p, \ (t_a \leq t_p), \ |\mathcal{S}_{ap}^<| > 0\}, \\
\mathcal{P}_+^> &= \{(a,p) \mid \mathrm{id}_a = \mathrm{id}_p, \ (t_a \geq t_p), \ |\mathcal{S}_{ap}^>| > 0\}.
\end{aligned}
\tag{3}
$$

The functional form of each per-pair term $\ell_{ap}^<$ and $\ell_{ap}^>$ follows the same probabilistic formulation as the Rank-N-Contrast (RnC) loss (Zha et al., 2023). ChronoCon differs, however, in how contrastive pairs and negatives are constructed: instead of relying on distances in label space, negatives are defined through temporal ordering within the same subject. To account for the asymmetry introduced by timestamps, the loss is further split into forward and backward chronological contributions. Furthermore a minor adjustment to the normalization is made, which is mainly relevant for short time series and small batch sizes.

Our loss provides a natural way of enforcing order via ranking with respect to $t$ for all samples sharing the same group identifier ($\mathrm{id}$). We explicitly avoid imposing a metric on $t$. Intuitively, such ordering should also support improved prediction of the target $y$ in downstream tasks, provided $y$ and $t$ exhibit a monotone relationship.

*Ordinal contrastive learning.* – This property also makes the loss well suited for ordinal regression. While not our primary focus, we evaluate the loss on ordinal disease-severity labels by adjusting the group identifiers accordingly. To distinguish this setting from our main objective—the unsupervised application to time-stamps—we denote the loss used on labels $y$ as $L^{\mathrm{OrdinalCon:Y}}$, emphasizing the *ordinality* in the pair selection process. [3]

**Data augmentation.** Without augmentation, only image series of length *three or more* would contribute to the loss, as at least one anchor, one positive, and one negative are required. To enable training on series with only two visits, we apply double-crop augmentation following (Zha et al., 2023).

**Application of ChronoCon in rheumatoid arthritis (RA) radiographs** We evaluate this approach on radiographs patients with RA to demonstrate that chronological information in routine imaging can yield clinically meaningful representations even without expert annotations. Disease severity in RA is commonly quantified using the Sharp–van der Heijde (SvH) score, which aggregates erosion (ERO) and joint-space narrowing (JSN) subscores for multiple joints, resulting in a total score ranging from 0 to 448. These subscores

---

2. One might also attempt to use only forward chronological contributions, but then early images would be under-represented as positives and late images over-represented as negatives.

3. One might in this respect refer to $L^{\mathrm{ChronoCon}}$ as $L^{\mathrm{OrdinalCon:t}}$, but we refrain from this to avoid confusion.

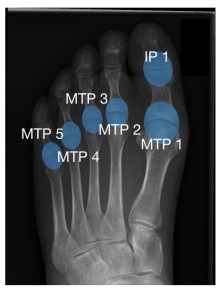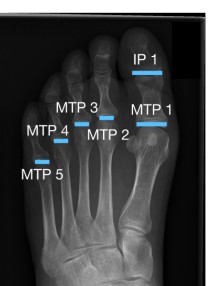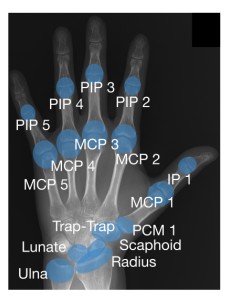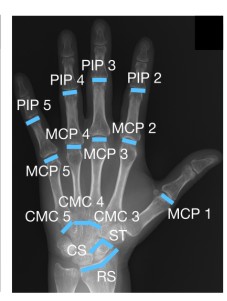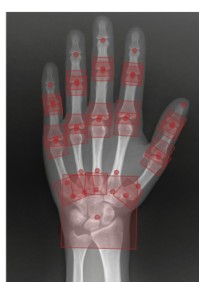

Figure 2: *Left:* Joint-level contributions to the total SvHS illustrated on a representative hand radiograph, highlighting erosions and joint spaces. *Right:* Regions of interest extracted during fully automatic preprocessing of hand radiographs.

are discrete and costly to obtain, making this domain a representative and challenging test case for label-efficient learning (van der Heijde, 2000).

First, we localize the joints with an automatic landmark-detection method (Payer et al., 2019; Jonkers et al., 2025) and extract an image patch for each detected joint (illustrated in Figure 2). For the first stage of training, contrastive pairs are constructed only from patches belonging to the *same patient, side, and joint type*. $\text{sim}(\mathbf{v}_i, \mathbf{v}_j) = -L_2(\mathbf{v}_i, \mathbf{v}_j)/\tau$ with temperature $\tau = 1$ was used as similarity metric throughout. To stabilize training, we add a standard denoising autoencoder (DAE) reconstruction loss $L_2^{\text{DAE}}$.

In the second stage, a multi-headed regressor replaces the decoder, with one head for each score type (59 in total). During fine-tuning, the encoder parameters are also updated, but with a reduced learning rate. The only loss used in this stage is mean-squared error (MSE) on the ERO/JSN scores.

All models are trained solely to predict *cross-sectional* SvH scores. Individual JSN and ERO scores are then summed to obtain the total SvH score, and differences between visits ($\Delta$SvHS) are computed afterward.

**Quality measures.** Performance is evaluated for both single–time point predictions and the derived progression $\Delta$SvHS. Agreement with ground truth is quantified using the intraclass correlation coefficient (ICC), root-mean squared error (RMSE), and Pearson's correlation coefficient $\rho$. For significance tests between models we always used a two-sided paired $t$-test on MSE (without the bootstrapping procedure). Full metric definitions and elaboration on performed statistics are in Appendix C.

## 3. Experiments and Results

**Dataset** The dataset consists of hand and foot radiographs from 778 patients with RA. It comprises 13 742 radiographic images and a total of 407 045 individual scores across 59 score types. As detailed in Table A in the appendix, the score distribution is highly imbalanced: fewer than 1% of erosion scores fall into the highest category, and fewer than 4% of JSN

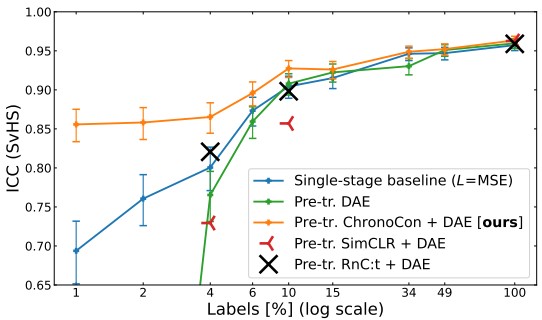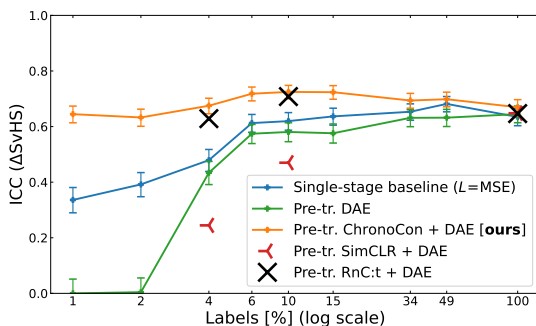

Figure 3: ICC of standard of reference and estimated SvHS as a function of training set size: (*left*) *SvHS*, and (*right*) change $\Delta SvHS$; blue: only single-stage baseline, green: pre-trained with reconstruction loss; orange: pre-trained with ChronoCon and reconstruction loss. Black cross: Pretrained with original Rank-N-Contrastive loss *on time*; ⊰ : pre-trained with SimCLR. Error bars indicate 95% CI.

scores do. The dataset also exhibits only short longitudinal series, with a median of 4 visits per patient (IQR [3,5]).

We use a patient-level split to avoid leakage of longitudinal information. Training, validation, and test sets contain $466/155/157$ patients with $8\,157/2\,753/2\,832$ images and $241\,701/81\,501/83\,843$ scores, respectively.

**Model and training details** We used ResNet18 as the encoder for all models (He et al., 2015). For DAE pretraining, the decoder mirrored the encoder using transposed convolutions. A hierarchically grouped dataloader was used to improve temporal consistency: patches from the same ROI and patient were typically placed in the same batch and oversampled based on intrapatient median ERO/JSN scores to mitigate score imbalance. Early stopping on validation mean absolute error (MAE) with a 10-epoch patience was applied for fine-tuning and for the supervised baseline, restoring the best-performing model.

All methods, except the single-stage baseline, follow a unified two-stage protocol introduced in the Methods section. In *Stage 1 (pretraining)*, the encoder is trained without labels using either a contrastive loss or a reconstruction loss. For contrastive methods, we denote the general loss as $L^{\mathrm{Con}} + 10^3\, L_2^{\mathrm{DAE}}$. where the prefactor scales the MSE-reconstruction loss to a similar order of magnitude as the contrastive loss.

The contrastive loss $L^{\mathrm{Con}}$ is instantiated in three variants: *ChronoCon*, which uses patient visit order to define positive and negative pairs; *RnC:t*, which is similar but defines negatives solely based on temporal distance, $\mathcal{S}_{ap}^{\mathrm{RnC:t}} := \{n \mid n \neq a,\ |t_a - t_n| \geq |t_a - t_p|\}$; and *SimCLR* (Chen et al., 2020), which uses standard contrastive learning without temporal or label information. DAE pretraining uses only the reconstruction loss $L_2^{\mathrm{DAE}}$. Double-crop augmentation is applied for all contrastive methods; for non-contrastive methods, the second crop is discarded. Experiments with an attached decoder (DAE variants) used half the batch size due to memory constraints.

| Dataset size | | | | Cross sect. (SvHS) | | Longit. (ΔSvHS) | |
|---|---|---|---|---|---|---|---|
| scores [%] | scores [$N$] | images | patients | RMSE ↓ | ICC ↑ | RMSE ↓ | ICC ↑ |
| 1 | 2 442 | 82 | 5 | $19.9 \, ^{-7.0}_{\pm 2.5}$ | $86 \, ^{+17}_{\pm 2}$ | $9.5 \, ^{-1.9}_{\pm 0.7}$ | $64 \, ^{+30}_{\pm 3}$ |
| 2 | 4 475 | 152 | 10 | $19.3 \, ^{-4.5}_{\pm 2.6}$ | $86 \, ^{+10}_{\pm 2}$ | $9.0 \, ^{-2.1}_{\pm 0.6}$ | $63 \, ^{+24}_{\pm 3}$ |
| 4 | 9 413 | 319 | 20 | $18.7 \, ^{-4.7}_{\pm 2.7}$ | $87 \, ^{+7}_{\pm 2}$ | $8.1 \, ^{-2.3}_{\pm 0.5}$ | $67 \, ^{+19}_{\pm 3}$ |
| 6 | 14 719 | 499 | 31 | $17.2 \, ^{-2.1}_{\pm 2.7}$ | $90 \, ^{+3}_{\pm 2}$ | $7.7 \, ^{-1.4}_{\pm 0.6}$ | $72 \, ^{+11}_{\pm 2}$ |
| 10 | 23 546 | 808 | 46 | $14.5 \, ^{-2.9}_{\pm 2.1}$ | $93 \, ^{+3}_{\pm 1}$ | $7.8 \, ^{-1.1}_{\pm 0.5}$ | $72 \, ^{+10}_{\pm 2}$ |
| 15 | 36 243 | 1 238 | 71 | $14.5 \, ^{-1.6}_{\pm 2.1}$ | $93 \, ^{+1}_{\pm 1}$ | $7.7 \, ^{-1.0}_{\pm 0.5}$ | $72 \, ^{+8}_{\pm 2}$ |
| 34 | 79 799 | 2 745 | 156 | $12.5 \, ^{-0.5}_{\pm 1.6}$ | $95 \, ^{+0}_{\pm 1}$ | $8.2 \, ^{-0.5}_{\pm 0.7}$ | $69 \, ^{+4}_{\pm 3}$ |
| 49 | 116 448 | 3 989 | 226 | $12.1 \, ^{-1.2}_{\pm 1.6}$ | $95 \, ^{+0}_{\pm 1}$ | $8.0 \, ^{-0.2}_{\pm 0.6}$ | $70 \, ^{+2}_{\pm 3}$ |
| 100 | 237 733 | 8 157 | 466 | $10.8 \, ^{-1.0}_{\pm 1.3}$ | $96 \, ^{+0}_{\pm 1}$ | $8.4 \, ^{-0.6}_{\pm 0.7}$ | $67 \, ^{+4}_{\pm 3}$ |

Table 1: Test-set performance of our model after pretraining with ChronoCon loss on 466 patients and fine-tuning on a fraction of labeled data. Superscripts in green/red show improvement over the single-stage baseline ; subscripts give half the 95% CI.

In *Stage 2 (fine-tuning)*, the decoder is replaced by a multi-headed regressor and the encoder is fine-tuned on available labels with MSE, using a learning rate reduced by a factor of 10. The single-stage baseline skips Stage 1 and trains encoder and regressor directly from ImageNet initialization. The code is publicly available at https://github.com/cirmuw/ChronoCon (further details in Appendix C).

### 3.1. Label efficiency

A key advantage of our loss is that it enables learning meaningful feature representations without access to scores $y$. To evaluate label efficiency, we created progressively smaller training subsets by reducing the number of patients with labeled data. The full training set comprises images from 466 patients. All splits were performed at the patient level, allowing us to simulate how performance changes when labels are available for only a subset of patients. The validation and test sets remained fixed across all experiments (155 and 157 patients, respectively). Details on the splits are provided in Table 1. Superscripts indicate improvements over the single-stage baseline (difference between orange and blue line in Figure 3.)

**Observations and Interpretation.** Our method using $L^{\text{ChronoCon}} + L_2^{\text{DAE}}$ outperforms the baseline across all label sub-splits. The improvement is driven by $L^{\text{ChronoCon}}$, not by $L_2^{\text{DAE}}$; in fact, DAE pretraining alone performs worse.

Changing the *ChronoCon* loss to *RnC:t* worsened the performance significantly in the low-label setting highlighting the importance of using a loss which allows for a non-linear

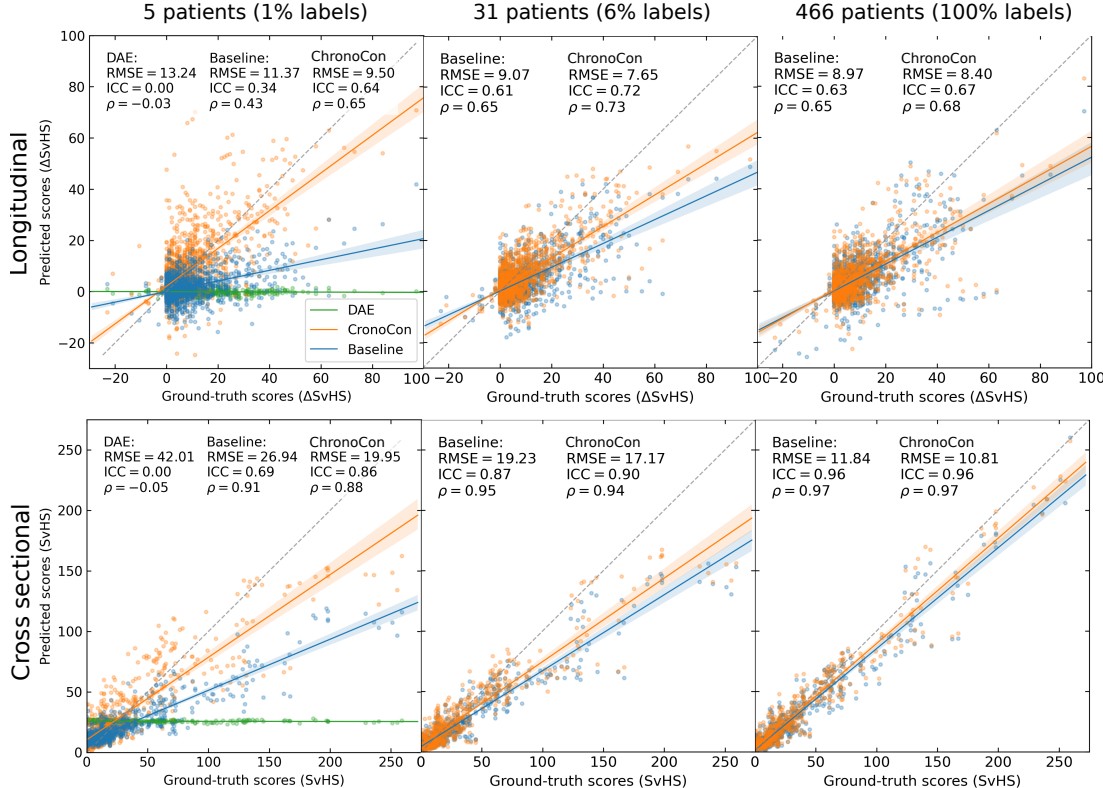

Figure 4: Scatter plots comparing ground truth and model predictions for the single-stage baseline (blue) and our ChronoCon method ($L^{\text{ChronoCon}} + L_2^{\text{DAE}}$; orange) trained on labels from 5, 31, and 466 patients. $L_2^{\text{DAE}}$ only results are shown in green. *Top:* Longitudinal evaluation in terms of score differences between visits. *Bottom:* Cross-sectional prediction performance for total SvH scores.

relationship between time and disease-progression. However, overall the performance of *RnC:t* was still good, likely because many of our image time-series consist of just a few images.

Performance gains are most pronounced in the low-label setting. Even in a *few-shot* scenario with labels from only 5 patients, our method achieves an ICC of 0.86 and RMSE of 19.9. For context, the SvHS has a standard deviation of 46 in the full dataset. Remarkably, ChronoCon trained on just 5 patients performs on par with a recently published model (RMSE = 23.6) trained on 367 patients [4]

Longitudinal evaluation ($\Delta$SvHS) shows even larger gains from ChronoCon in the low-label regime. Notably, its performance remains almost constant over a wide range of training-set sizes, unlike the baseline. This stems from explicitly learning patient-specific progression

---

4. and substantially outperforms (Moradmand and Ren, 2025a) (trained on 428 patients/visits) who reported RMSE = 44.28 on scores 0–270.

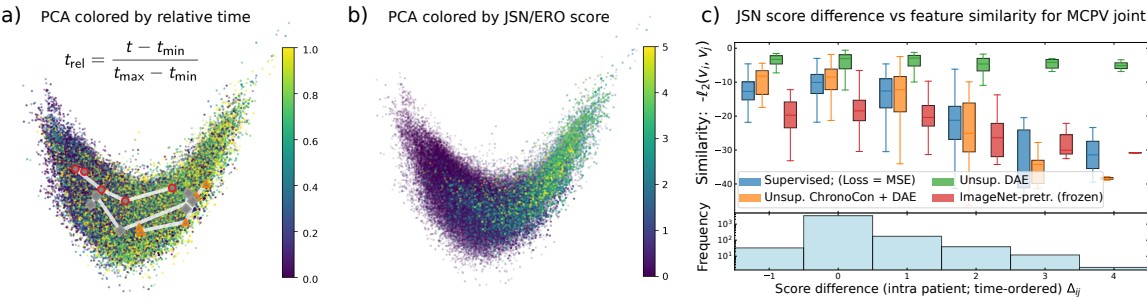

Figure 5: Feature space (PCA) of the unsupervised model pretrained with ChronoCon. *Left (a):* Colored by relative time $t_{\mathrm{rel}}$ ($0$ = first visit, $1$ = last); white lines show example patient–joint trajectories. *Middle (b):* Same embedding colored by ground-truth scores (no score information was used during training). *Right (c):* Feature similarity between chronologically ordered visits compared to the corresponding joint-space–narrowing (JSN) score differences for the MCPV joint.

in an unsupervised manner. With only 4% of labels, results match those obtained using 100% of labels. Interestingly, longitudinal performance peaks at 6–15% of labels rather than at full supervision, suggesting that strong (and noisy) labels may override the features learned during pretraining.

*Scoring consistency*—The progression error are substantially better than expected from subtracting two noisy estimates. If $\hat{y}_i = y_i + \epsilon_i$ with noise variance $\sigma^2 = \mathbb{E}[\epsilon_i^2]$, then $\mathrm{MSE}(\Delta\mathrm{SvHS}) = 2\sigma^2(1 - c)$ , where $\mathbb{E}[\epsilon_i\epsilon_j]/\sigma^2 = c$ is the error correlation ($i \neq j$). For uncorrelated errors, $\mathrm{RMSE}(\Delta\mathrm{SvHS})$ should be $\sqrt{2}$ times $\mathrm{RMSE}(\mathrm{SvHS})$. However, our errors are highly correlated ($c = 0.91$ for 4% labels; $c = 0.70$ for 100%), indicating strong error cancellation—i.e., *scoring consistency*—at least partly due to ChronoCon pretraining.

### 3.2. Learned feature space

To further investigate the feature space learned in the first stage with ChronoCon, the embedding is visualized in Figure 5 *a* and *b*. The 512-dimensional features were reduced to 2 dimensions using principal component analysis (PCA). The training process is completely invariant to global time shifts for each patient because only repeatedly acquired patches of the same patient, side, and joint type are contrasted against each other. In Figure 5 *a*, the embedding is colored by relative time, with three example trajectories shown as white lines. In Figure 5 *b* the same embedding is colored by JSN/ERO labels. Importantly, no labels were used during training. Points are displayed in order of increasing score to highlight the transition from low to high severity. All ERO and JSN patches are shown in the same plot, even though their respective maximum scores differ (5 and 4).

In Figure 5 *c* we compare pairwise feature similarities over disease-label differences for the MCPV joint. Only features for the same patient and side are compared. Equivalent plots for all other joints are provided in the appendix (Figure 6). The visit pairs are chronologically

ordered, though not necessarily consecutive for patients with more than two visits. The lower panel displays a histogram of disease-label- (JSN-score) -differences between the two visits. The histogram shows that while scores typically increase over time, decreases of up to -1 also occur, likely due to quantization effects in the discrete scoring system.

The DAE baseline is shown in green, our main model—which combines the reconstruction loss with ChronoCon—is shown in orange, and the untrained/frozen ResNet18 (initialized from ImageNet-weights) is shown in red. For the latter the image-net classification layer was removed; leaving a total of 17 layers from the ResNet18. None of these three models had access to ERO/JSN scores. The only model which used scores during training is the single-stage baseline shown in blue.

**Observations and interpretation.** Among the unsupervised baselines, the DAE shows the weakest behavior: its feature similarities exhibit little correspondence with progression severity. In contrast, the frozen ImageNet encoder can already distinguish progression to some extent—the red box-plots roughly follow the trend of the supervised baseline. This aligns with the observation that the supervised baseline, when trained on labels from only 5 patients, still achieves an ICC of 0.34 for $\Delta$SvHS.

Our full model provides the clearest separation of progression patterns. The embeddings are effectively ordered only along each *individual* trajectory and disease-related features are learned automatically. Remarkably, coloring the embedding by visit time appears more disordered than when coloring by severity. This suggests that, in the second stage, the regression heads only need to learn which regions of feature space correspond to which scores—explaining the strong performance even with labels from just 5 patients. Consistently, the feature similarities of our model (orange) closely follow those of the supervised baseline (blue), which was trained on patch–score pairs from all 466 patients.

### 3.3. Combination with other pre-training methods and ablation study

Different pre-training strategies are summarized in Table 2. All models were trained on the full set of 466 patients and their corresponding scores. The table is organized into four groups: (i) the supervised baseline (top), (ii) unsupervised methods where no labels $y$ are used in the first stage, (iii) supervised first-stage training with the original RnC loss applied to labels, and (iv) supervised first-stage training with our loss applied to *labels* ($L^{\text{OrdinalCon:Y}}$).

Whenever a contrastive loss is applied to labels, stratification via `id` uses the score type (e.g., `IP_JSN`, `PIPII_ERO`, ... ), so any–vs–any patient pairs are allowed as long as they correspond to the same ROI and score type.

**Observations and interpretation.** Table 2 should be read by separating three regimes: (i) no labels in stage one, (ii) full-label pretraining, and (iii) time-order based pretraining.

In the no-labels pre-training setting, neither as a standalone task nor in combination with other losses did the reconstruction/denoising objective $L_2^{\text{DAE}}$ yield meaningful improvements. In contrast, ChronoCon pretraining improved performance—particularly ICC($\Delta$SvH)—over other methods that do not use label information. Regarding MSE, there is a statistically significant difference between *ChronoCon + DAE* and the single-stage baseline ($p < 10^{-4}$ cross-sectionally and longitudinally).

| | pre-training scheme | Cross sectional (SvHS) | | Longitudinal (ΔSvHS) | |
|---|---|---|---|---|---|
| | | RMSE ↓ | ICC ↑ | RMSE ↓ | ICC ↑ |
| | single stage baseline | 11.8 ± 1.4 | 95.7 ± 0.6 | 9.0 ± 0.8 | 63.5 ± 3.1 |
| no-label pretraining | $L_2^{DAE}$ | 12.9 ± 1.1 | 96.0 ± 0.6 | 9.1 ± 0.6 | 64.5 ± 3.0 |
| | $L^{RnC:t} + L_2^{DAE}$ | 11.4 ± 1.4 | 95.9 ± 0.6 | 8.7 ± 0.8 | 64.7 ± 3.0 |
| | $L^{SimCLR} + L_2^{DAE}$ | 11.1 ± 1.2 | 96.3 ± 0.6 | 8.9 ± 0.7 | 64.8 ± 3.0 |
| | $L^{ChronoCon}$ | 11.0 ± 1.3 | 96.4 ± 0.5 | 8.4 ± 0.7 | 66.8 ± 2.8 |
| | $L^{ChronoCon} + L_2^{DAE}$ | 10.8 ± 1.3 | 96.3 ± 0.6 | 8.4 ± 0.7 | 67.0 ± 2.8 |
| label pretraining | $L^{RnC} + L^{ChronoCon}$ | 10.9 ± 1.2 | 96.3 ± 0.6 | 8.7 ± 0.7 | 65.1 ± 3.0 |
| | $L^{RnC} + L^{ChronoCon} + L_2^{DAE}$ | 10.8 ± 1.2 | 96.3 ± 0.6 | 8.6 ± 0.7 | 66.7 ± 2.8 |
| | $L^{RnC}$ | 10.4 ± 1.1 | 96.6 ± 0.5 | 8.4 ± 0.7 | 67.6 ± 2.8 |
| | $L^{OrdinalCon:Y} + L^{ChronoCon}$ | 10.7 ± 1.3 | 96.4 ± 0.6 | 8.3 ± 0.7 | 68.4 ± 2.7 |
| | $L^{OrdinalCon:Y} + L^{ChronoCon} + L_2^{DAE}$ | 10.4 ± 1.1 | 96.6 ± 0.5 | 8.5 ± 0.7 | 67.1 ± 2.8 |
| | $L^{OrdinalCon:Y}$ | **9.8** ± 0.9 | **97.1** ± 0.4 | **8.2** ± 0.6 | **70.6** ± 2.6 |

Table 2: Different pre-training strategies on the full training set of 466 patients. Values are shown with 95% CI. Underlined metrics indicate the best methods that do not require labels during pretraining; bold values indicate the best overall performance. Except for the baseline, all models were trained in two stages. ⋆ / ∗ indicate cross-sectional / longitudinal results of a two-sided paired $t$-test on MSE. *Note:* $p$-values are reported for the individual paired comparisons described in the text; see Appendix for details and interpretation.

When all labels are used during pretraining (via $L^{OrdinalCon:Y}$ or $L^{RnC}$), these methods perform best. In this regime, there is no significant difference between *RnC* and *ChronoCon*. This is expected, as the supervision signal is already maximal. However, applying $L^{OrdinalCon:Y}$ to the ordinal JSN/ERO scores yields the strongest overall results, improving cross-sectional MSE over RnC ($p = 0.016$) We attribute this the the fact that when our loss is applied to labels ($L^{OrdinalCon:Y}$) it respects the ordinal structure of the labels ($0 \to 1 \neq 1 \to 2$), whereas the original RnC loss does not.

For time-order based pretraining, the *RnC:t* baseline performs slightly worse than *ChronoCon*, highlighting the advantage of separating positive- and negative-time directions. Similarly, the *SimCLR + DAE* baseline shows numerical differences to *ChronoCon + DAE* only in the longitudinal setting ($p = 0.0016$), indicating that visitation-order aware pretraining primarily benefits longitudinal metrics.

Importantly, Table 2 also shows that when labels are abundant, pretraining on visitation time does not add benefit over label-based pretraining. The main advantage of *ChronoCon* therefore lies in *low-label settings*, where time-order information substitutes for missing annotations.

## 4. Discussion

**Summary**   We introduced ChronoCon, a chronological contrastive loss that exploits visitation order in longitudinal imaging to learn disease-relevant representations without expert labels. In RA radiographs, the learned feature space captured both cross-sectional severity and longitudinal progression, and clearly improved performance in low-label scenarios compared with purely supervised or reconstruction-based pretraining. These findings highlight that chronological information routinely present in clinical archives can serve as a powerful and inexpensive supervisory signal.

Compared with typical representation-learning approaches that operate on individual images or unordered pairs, our method leverages temporal ordering as an additional inductive bias. While several self-supervised objectives have been explored in medical imaging, few explicitly account for longitudinal structure. The observed improvements in low-label settings suggest that temporal ordering can provide complementary information in an irreversible disease setting.

**Limitations and ethical concerns.**   The usefulness of ChronoCon depends on the presence of a valid ordering variable $t$ within subgroups of shared id. When $t$ denotes visit time, the loss assumes a predominantly monotonic progression. This is a reasonable approximation for erosive changes in RA (van der Heijde, 2000), but may not hold in diseases with non-monotonic or treatment-reversible patterns.

Beyond this conceptual limitation, several practical aspects should be noted. First, our experiments are based on a single-center dataset, and broader multi-center validation will be necessary. Second, the method relies on sufficient longitudinal coverage; in datasets dominated by single visitations, the benefit of chronological contrastive learning is limited.

Furthermore, the choice of subgroup identifiers also warrants careful consideration. In our RA study, id was defined at the joint level to avoid semantically implausible comparisons. More broadly, subgrouping can encode clinically meaningful structure, but our approach could be used with demographic or biologically sensitive categories which raises ethical concerns. Depending on the application, subgroup definitions may either improve representation quality or inadvertently entrench biases, making transparent justification essential.

**Conclusions.**   Chronological contrastive learning provides a simple and effective way to leverage unlabeled longitudinal imaging data for representation learning. By using only visitation order, it generalizes label-based contrastive ranking to a setting where expert scores are not required, enabling strong performance even when labels are scarce. Our experiments on RA radiographs demonstrate that chronological signals embedded in routine clinical workflows contain exploitable structure for learning progression-aware feature spaces. The approach has potential relevance for other predominantly irreversible diseases and may help reduce annotation burden in domains where expert scoring is costly or inconsistent.

## Acknowledgments

This project has been partially funded by: The Innovative Health Initiative Joint Undertaking (IHI JU) and its members, and other contributing partners, under grant agreement No. 101194766, the Vienna Science and Technology Fund (WWTF, PREDICTOME) [10.47379/LS20065], and the Austrian Science Fund (FWF, P35189). Some authors (C.W.) were supported by the Clinical Research Group MOTION, Medical University of Vienna, Vienna, Austria – a project funded by the Clinical Research Groups Program of the Ludwig Boltzmann Gesellschaft (Grant Nr: LBG_KFG_22_32) with funds from the Fonds Zukunft Österreich.
Co-Funded by the European Union, the private members, those contributing partners of the IHI JU, and SERI. Views and opinions expressed are, however, those of the authors only and do not necessarily reflect those of the aforementioned parties. Neither of the aforementioned parties can be held responsible for them.
C.W. thanks Marlene Steiner and Simon Schürer-Waldheim for many insightful discussions.

**Data Availability**   The data used in this study were obtained from an internal dataset of the Medical University of Vienna, collected within the AutoPIX consortium. Due to ethical, legal, and data protection constraints, the data are not publicly available. Access may be granted upon reasonable request and subject to institutional approval and appropriate data sharing agreements.

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

## Appendix A. Vienna RA dataset details

Data were collected between 1997 and February 2018 at the Division of Rheumatology, Department of Internal Medicine III, Medical University of Vienna. The study protocol

| Type | Score distribution (counts) | | | | | | |
|---|---|---|---|---|---|---|---|
| | 0 | 1 | 2 | 3 | 4 | 5 | NS |
| ERO hands | 160 000 | 11 838 | 3 657 | 1 603 | 395 | 412 | 2 613 |
| ERO feet | 66 581 | 8 733 | 2 472 | 980 | 221 | 424 | 2 177 |
| JSN hands | 63 165 | 24 048 | 7 043 | 5 248 | 3 093 | – | 1 548 |
| JSN feet | 24 279 | 9 221 | 2 338 | 2 225 | 1 640 | – | 1 091 |

Table 3: Erosion and joint-space narrowing scores on joint-(part) level. NS = Not scoreable (e.g. surgical spacers, fused joints, missing fingers, ...).

outlining the retrospective data analysis was approved by the local ethical committee of the Medical University of Vienna (vote number : 1206/2018). Table A shows the score distribution for erosion and joint-space narrowing speperated by hands and feet joints. More information on the dataset can also be found in (Deimel et al., 2025).

## Appendix B. Related work for RA and SvH score estimation

Most published work on SvHS prediction in RA has relied on fully supervised learning without unsupervised pretraining beyond ImageNet initialization (Sun et al., 2022). (Maziarz et al., 2022) a combined objective of ROI segmentation together and smoothed label classification highlighting to address the quantification error in JSN and ERO scores. (Bo et al., 2025) proposed an attention-based multiple-instance learning model to obtain an interpretable SvHS predictor. (Moradmand and Ren, 2025b) used a vision transformer to aggregate per-joint predictions into a total SvH score, achieving strong performance in common, less severe cases. The winning RA2-DREAM Challenge approach (Team, 2020) used a pipeline that combined joint localization through segmentation with a subsequent model that integrates all joint scores per patient. More recent work explored self-supervision (Ling et al., 2024) and unsupervised pretraining of a vision transformer in a cohort of patients with psoriatic arthritis (Govind et al., 2025). To our knowledge, no existing model explicitly leverages the time-series structure of longitudinal RA imaging.

## Appendix C. Training-/ evaluation details and additional results

Training settings and code of ChronoCon is available at https://github.com/cirmuw/ChronoCon. Our point-annotation tool and the landmark detection code for the pre-processing steps is in a separate repository https://github.com/cirmuw/autopix_landmarks_utils.

**Preprocessing** Preprocessing followed the pipeline described in (Deimel et al., 2025). All right-hand and right-foot radiographs were horizontally mirrored for consistency. Images originally encoded in the DICOM MONOCHROME2 format (black foreground on white background) were converted to MONOCHROME1. Radiographs containing both hands or both feet were split at the midline.

**Joint detection**    Joint localization was performed using the Spatial Configuration Network (SCN) (Payer et al., 2019), implemented as described in (Jonkers et al., 2025). After training on 480 radiographs, landmark detection achieved a mean median point-to-point error of $<1.0\,\mathrm{mm}$ (mean over ROIs; median over samples) on a test set of 40 radiographs. Detailed results are available online (`https://github.com/cirmuw/autopix_landmarks_utils/tree/main/nb/landmarks_evaluation`). Square patches of size $156 \times 156$ pixels were extracted around each region of interest (ROI).

**Data augmentation**    Data augmentation included random rotations (up to 10°), translations (up to 17 pixels), followed by a center crop to $128 \times 128$ pixels to avoid padded boundaries. Photometric augmentations consisted of random intensity scaling, intensity shifting, contrast adjustment, and histogram shifting. Additional robustness augmentations included Gaussian smoothing and light noise perturbations. All image patches were finally normalized to the intensity range $[0, 1]$.

**Encoder**    All experiments used a ResNet18 encoder. Prior to training, the encoder was initialized with ImageNet-pretrained weights.

**Decoder**    For reconstruction or denoising tasks, a decoder mirroring the ResNet18 architecture was employed, with deconvolution (transposed convolution) layers replacing convolutional downsampling layers. When reconstruction was used, Gaussian noise of magnitude $10^{-5}$ (with clip) was added to the input to implement a denoising autoencoder. Whenever the decoder was included, memory requirements doubled and the batch size was therefore halved.

**Regression heads**    Score prediction used a multi-headed regression module comprising 59 independent heads (one per score subtype). Each head was a multilayer perceptron with two hidden layers of dimension 128. Whenever supervised regression was used, the MSE loss was applied.

**Score summation and extrapolation**    For erosion, the proximal and distal parts of the affected joints were scored separately. In the computation of the total SvH score, these two parts were summed, and the model predictions were combined in the same way. For foot joints, the total erosion score per joint must not exceed 10 (5 for each joint part). Accordingly, the outputs of the regression heads ($y \in \mathbb{R}$) were clipped to the range $[0, 5]$ for each foot joint part. For the PIP and MCP joints of the hand, the sum of proximal and distal parts was clipped to the range $[0, 5]$.

**Metrics and analysis**    Erosion and joint-space-narrowing scores were summed per visit to obtain the total SvH score. When subscores were missing (e.g., not scoreable due to surgery), the total score was estimated via linear interpolation. Visits with more than 25% missing subscores were excluded. To assess the model's ability to capture progression, we evaluated the change in total SvH score between visits, $\Delta\mathrm{SvHS} = \mathrm{SvHS}(t_2) - \mathrm{SvHS}(t_1)$.

Intraclass correlation coefficients were computed using a two-way mixed-effects model with single measures and absolute agreement (ICC3-1 in the terminology of (Shrout and Fleiss, 1979)). All ICC values and confidence intervals were calculated with the $R$ package `psych` (R Core Team, 2021; William Revelle, 2025).

In scatter plots of true vs. predicted SvH or $\Delta$SvH scores, Pearson's correlation coefficient $\rho$ is reported. Error bars represent 95% confidence intervals (CI). For both the root-mean-squared error (RMSE) and $\rho$, confidence intervals were obtained via bootstrapping. In tables, reported $\pm$ values correspond to half the width of the 95% CI.

**Training parameters** Models were trained on NVIDIA A100 GPUs (40 GB VRAM) using the AdamW optimizer. Batch sizes were 512 without a decoder and 256 with a decoder, with learning rates scaled proportionally for smaller batches. A `ReduceLROnPlateau` scheduler was used to reduce the learning rate when the validation loss plateaued.

**Hyperparameter search** Learning rates and the contrastive temperature $\tau$ were optimized using `optuna`'s TPE sampler (Bergstra et al., 2011) on the PIPIII and MCPIII joints (six score types) from the 466 training patients. Search ranges were $\tau \in [0.1, 5]$, encoder LR $\in [10^{-7}, 10^{-2}]$, head LR $\in [10^{-5}, 10^{-2}]$, and weight decay $\in [10^{-8}, 10^{-1}]$. The search yielded $\tau = 1$ as temperature (prefactor to $L_2$ feature similarity), an encoder LR of $4 \cdot 10^{-4}$, a head LR of $4 \cdot 10^{-5}$, and a weight decay of $10^{-6}$ for a batch size of 512, with proportional LR scaling for smaller batches.

**SimCLR parameters** Parameters and settings for the SimCLR baseline were taken directly from the original publication (Chen et al., 2020) without further hyper-parameter search. (Added in the rebuttal). MLP projector with a single hidden layer and an output-dimension of 128; Temperature $\tau = 0.07$; Feature-similarity: cosine;

**Statistical analysis** All hypothesis tests were performed on paired, per-instance MSE differences on the fixed test set (`scipy.stats.ttest_rel(..., alternative='two-sided')`). Bootstrapping was used only for confidence intervals of ICC and RMSE and was not involved in hypothesis testing.

In contrast to ICC/RMSE, where there is a nonlinear relationship between the statistic and the sample estimations, not bootstrapping is used for the statistical tests of MSE.

Several paired hypothesis tests are reported in Table 2. These tests are intended to provide quantitative support for observed performance differences between specific model pairs, rather than to establish confirmatory claims across a family of hypotheses. The reported $p$-values should therefore be interpreted in this descriptive context.

| Abbreviation | Meaning |
|---|---|
| RA | Rheumatoid Arthritis |
| SvH / SvHS | Sharp–van der Heijde Score |
| $\Delta$SvHS | Change in SvH score between visits (in chronological order) |
| ERO | Erosion |
| JSN | Joint Space Narrowing |
| ROI | Region of Interest |
| NS | Not Scoreable |
| ICC | Intraclass Correlation Coefficient |
| RMSE | Root-Mean-Squared Error |
| MAE | Mean Absolute Error |
| DAE | Denoising Autoencoder |
| RNC | Rank-N-Contrast (loss) |
| ChronoCon | Chronological Contrastive Loss (this work) |
| SCN | Spatial Configuration Network |
| TPE | Tree-Structured Parzen Estimator (Optuna) |

Table 4: List of abbreviations beyond used throughout the manuscript and appendix.

| Abbreviation | Description |
|---|---|
| **Hand joints** | |
| PIPII–PIPV | Proximal interphalangeal joints II–V (JSN) |
| PIPIIED/EP ... VED/EP | PIP joints II–V, distal/proximal erosion |
| MCPI–MCPV | Metacarpophalangeal joints I–V (JSN) |
| MCPIED/EP ... VED/EP | MCP joints I–V, distal/proximal erosion |
| IPIED / IPIEP | Thumb IP joint distal/proximal erosion |
| Rad_Carp | Radiocarpal joint (JSN) |
| RadiusE, UlnaE | Radius / Ulna erosion |
| LunatE, ScaphE, TrapE | Carpal bone erosions (lunate, scaphoid, trapezium) |
| Sca_Cap, Tra_Sca | Carpal articulation JSN |
| Base_MCIE | Base of metacarpal I (erosion) |
| **Foot joints** | |
| MTPI–MTPV | Metatarsophalangeal joints I–V (JSN) |
| MTPIED/EP ... VED/EP | MTP joints I–V, distal/proximal erosion |
| IP | Hallux interphalangeal joint (JSN) |
| IPED / IPEP | Hallux interphalangeal distal/proximal erosion |
| `<joint>_ED` | Distal joint part (erosion) |
| `<joint>_EP` | Proximal joint part (erosion) |

Table 5: Grouped joint and score abbreviations used in this work contributing to Sharp-van der Heijde score . Erosion applies separately to proximal (EP) and distal (ED) joint parts.

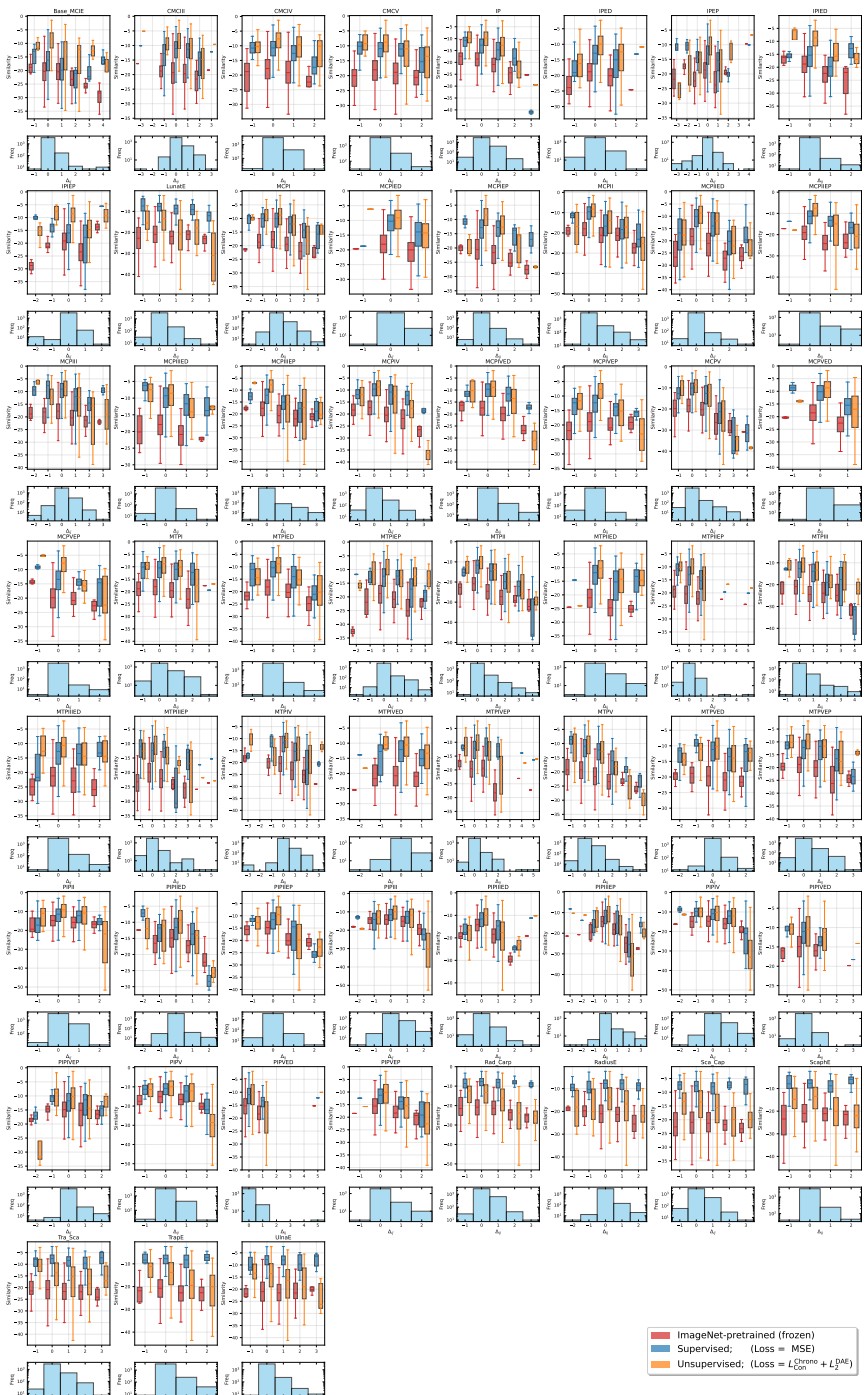

Figure 6: Feature similarity $(-L_2(\mathbf{v}_i, \mathbf{v}_j))$ between different chronologically ordered visits compared to the difference in score (ground truths).

