# OpenReview forum: "Chronological Contrastive Learning: Few-Shot Progression Assessment in Irreversible Diseases"
_MIDL.io/2026/Conference — MIDL 2026 Poster_

### Official Review · Reviewer_9ysM · 2026-01-09

**Confidence:** 3
**Preliminary Rating:** 3
**Final Rating:** 5

**Summary:**

The authors present ChronoCon, a contrastive pretraining strategy based on longitudinal unlabeled image data for encoder (and encoder-decoder) architectures with the aim to increase cross-sectional disease severity prediction. The work uses the RnC loss with an adjusted definition of negatives that intuitively fits the progression dynamics of monotonically progressing diseases. Instead of relying on expert-guidance (e.g., RnC uses disease labels) or visit times, the architecture only uses the temporal order of screenings in a subject's longitude. It is applied to an in-house dataset with rheumatoid arthritis scores for multiple joints per foot or hand radiograph, and performs on par with the RnC baseline and better than the fully supervised same-encoder baseline in predicting (consistent) severity scores. The performance increase compared to the fully supervised baseline is higher in the low data regime, making it significantly more data efficient. If, instead of screening order, disease labels are used during contrastive pretraining, matching the RnC baseline's input, the model outperforms the RnC. The learned embeddings follow the disease severity order.

**Strengths:**

This work investigates changes to the definition of negatives, among other things, of the RnC loss to improve progression-aware contrastive pretraining on longitudinal data. This is of potential value for many medical imaging problems that predict disease severity or prognose progression dynamics. The proposed RnC-based loss captures natural disease progression of monotonously worsening diseases well and this notion of disease dynamic formulation constitutes a research gap that the authors fill. The loss can flexibly be added to any encoder or encoder-decoder architectures and not needing any expert guidance allows leveraging longitudinal unlabeled data that comes with absolutely minimal information. The study shows that the proposed loss, which is related to RnC, performs on par with RnC or outperforms it on in-house data, depending on the metadata used (temporal order or disease label). The progression-aware pretraining was shown to be beneficial compared to full finetuning, especially when dealing with few training data, which is frequently the case in medical imaging. The manuscript presents solid work, a good choice of baselines, ablation and, metrics.

**Weaknesses:**

Despite the manuscript being written well in terms of language and detail, it partly lacks consistent structure and polish. Furthermore, the comparison baselines lack an obvious variant. Also, the dataset is highly specific, featuring many scores per image region, which makes it hard to judge the generalization of the proposed method to more typical datasets.

**Detailed Comments:**

- The advantage of ChronoCon vs. an RnC variant with temporal order or visit time as y-label is unclear as it had not been compared. The time of screening is arguably abundantly available for a scan, even if no other expert annotations are available. For that reason, I would have expected the authors to compare against RnC with screening time and/or temporal order instead of y, too. Do you agree that $L^{RNC:t}$ would be a worthwhile addition?
- The relation to RnC is not fully transparent. Do you essentially (aside from the decoder and denoising loss) adjust the RnC loss by changing the definition of negatives? If so, please state that explicitly. This way, it would also be clear that the loss terms in Eq. 3 are the per-pair loss term from RnC (just below Eq. 1).
- While potentially not feasible in this study, the application to a less specific dataset (that needs no region-specific crops, etc.), would be interesting.
- The mentioned related works are a reasonable subset of the body of literature on this topic. I wonder, however, how does this work relate to or diverge from Kim & Sabuncu (2023_learning) who make use of the temporal order, too? [Do you consider any of the following works related enough? Holland et al. (2024_metadata), Chakravarty et al. (2024_predicting), Zeghlache et al. (2024_latim)]
- Dataset specifics are mentioned at various places, making it unnecessarily challenging to follow. For instance, content that should go in "Application of ChronoCon in RA" are written in "Quality measures". I would appreciate a more compiled/collective presentation of information that explains how the application and data interact. Also a general training procedure is described under "Label Efficiency", which is confusing. The subscript and superscript inconsistency of loss variants is confusing (E.g., "Con" can be the subscript vs. "2", where the first has no particular meaning other than being the second part of the architecture name, and the latter has). The differentiation between the fully supervised baseline (termed "single stage baseline") and expert-guided contrastive pretraining (termed "supervised") could be improved. The S^RnC formula has a "(" too much at p. 3, top.
- Overall the document structure is not entirely typical, which can be fine, but maybe also is a source of the above mentioned structure issues.
- Did the fully supervised baseline and RnC variants receive the double crop augmentations, too, and with that, the same batch size (except in case of decoder variants)?
- Does $\Delta$SvHs consider the difference in predicted aggregate score of two consecutive screenings?
- The 17-Layer Resnet needs motivational introduction. Overall, Fig. 5 and the corresponding text would benefit from an introductory sentence stating what to expect here.
- The reproduced figure should have a short permission statement attached.
- Tab. 2 is referred to as Tab. 3.3
- Dataset availability information is missing.
- The code and weights are not published yet. Since publishing these affects the paper's impact within the community, I would argue that promising its release post acceptance is not sufficient for the reviewers to fully judge the work, and encourage you to provide code sooner.

**Justification Of Final Rating:**

The authors provided a substantial rebuttal and discussion and this improved the manuscript greatly. The added clarity and additional comparison, as well as the clarification answers by the authors make me suggest the acceptance of this work with a rating of 4 or 5, and, given the extent of the revision, choosing 5.

**Justification Of The Preliminary Rating:**

Overall, the manuscript is not yet concisely structured and polished, and some clarification or addition of experiments are needed. I am confident that addressing the comments can yield a good manuscript.

**Questions To Address In The Rebuttal:**

All of the above.

---

> ### Author Response · Authors · 2026-01-24
> **RnC:t comparison, clarified loss formulation, and strengthened presentation and evaluation**
>
> We would like to thank the referee for the careful evaluation of our paper, for remarking on its general applicability, the significant improvements regarding data efficiency and especially for the detailed and constructive feedback.
>
> In the following we respond to each of these comments case by case. Note that we abbreviated most of the referee's comments to bullet points.
>
> ---
>
>
> > RnC with temporal order (*RnC:t*)
>
>
> We agree that this baseline/ablation (*RnC:t*) makes for a workwhile addition to the manuscript. During the rebuttal we performed additional experiments for this setting (See figure 3; table 2). Our method (ChronoCon) shows higher label-efficiency compared to this baseline.
>
> However, while ChronoCon also shows better performance than RnC:t in the setting where all samples come with expert labels (table 2), this difference is not statistically significant. This is not surprising, given that our method's main benefit is label efficiency. In the low-label setting the difference is statistically significant (e.g. $p<10^{-6}$ cross-sectionally and longitudinally for 4% of labels).
>
> > The relation to RnC is not fully transparent. Do you essentially (aside from the decoder and denoising loss) adjust the RnC loss by changing the definition of negatives? If so, please state that explicitly. This way, it would also be clear that the loss terms in Eq. 3 are the per-pair loss term from RnC (just below Eq. 1).
>
> Yes. The main difference is the definition of negatives, which requires splitting the loss into two parts to avoid temporal bias. We also normalize only over valid (a,p) pairs with existing negatives, similar in spirit to triplet loss normalization. We now clarified this relation in the manuscript, below after Eq. 3.
>
>
> > Application to less specific datasets
>
> We acknowledge that this is one of the paper’s weaknesses. While we have plans for applying the RnC loss to other longitudinal datasets in the near future, we must agree with the referee that it is unfeasible in this study.
>
>
> > Relation to Kim, Holland, Chakravarty, Zeghlache
>
> We thank the referee for pointing us to these works and added them to the related work section. The differences and similarities are:
>
> *[Kim & Sabuncu (MIDL 2023)]* also assume monotonic progression but operate on image pairs with a learned classifier, whereas ChronoCon uses full trajectories and a non-parametric latent loss. They also assume monotone progression, but their approach further differs in the prediction target. Specifically, they predict the difference between images directly, whereas we attempt to predict in a cross-sectional manner with sufficient precision and consistency that also the longitudinal assessment is accurate. This poses a harder problem.
>
> *[Holland et al. (MIA 2024)]* define positives within a fixed time window and negatives across patients, which assumes known progression timescales, frequent acquisitions, and balanced disease states (randomly sampled negatives are assumed to be in a different disease state). These assumptions are unlikely to hold in RA.
>
> *[Chakravarty et al. (MICCAI 2024)]* enforce temporal ordering via pairwise losses and parallel hyperplanes, again using image pairs rather than full sequences.
>
> *[Zeghlache et al. (MICCAI 2024)]* model continuous dynamics with Neural ODEs, which is more general but assumes differentiable feature evolution and adds ODE training complexity. ChronoCon is simpler and may be advantageous when abrupt, non-differentiable changes occur (e.g., surgery).
>
> > Structure, notation, terminology.
>
> A more compiled/collective presentation of information that explains how the application and data interact: We revised the manuscript structure, notation, and terminology accordingly. In particular we disentangled the general shared training setup from the specific experiments. See latexdiff.
>
> > Augmentations and batch size:
>
> All methods used the same augmentations and batch size. For the supervised baseline, the second crop was discarded. Decoder variants used half the batch size due to memory limits. We now state this explicitly.
>
> > Does SvHs consider the difference in predicted aggregate score of two consecutive screenings?
>
> SvHS is defined per image. However, the change between screenings (ΔSvHS) is a standard outcome measure in RA trials (Landewé et al., RMD Open, 2023).
>
>
> > ResNet motivation, figure, tables, data, code:
>
> We clarified the ResNet motivation, improved Fig. 5 introduction, replaced the reproduced figure with an original illustration, corrected table references, added dataset availability, and made the code public https://github.com/cirmuw/ChronoCon).
>
> ---
>
> Given these changes, which directly address the concerns and suggestions raised in the review, we hope that the referee finds that the revised manuscript now meets the standards for acceptance and we would be grateful if the rating could be reconsidered in light of the revised version.

---

> > ### Comment · Reviewer_9ysM · 2026-01-28
> >
> > The authors revised the manuscript, significantly improving clarity, detail, adding the proposed additional baseline, adding a code repo, and more. A few things should still be considered regarding new aspects introduced in the revised version, see below. I appreciate the attached diff pdf.
> >
> > - The authors added hypothesis testing to quantify performance gains. I appreciate that, however, a few points should be considered here.
> >     - (a) t-tests for a model comparison on the same testing set have been shown to underestimate variance and can lead to inflated type 1 error, and are therefore often considered misused in this context from a statistics perspective ([see e.g. this reference](https://doi.org/10.1038/s41598-024-56706-x)). I acknowledge, that, empirically, they are widely used despite these limitations, and the referenced study argues that misuse does not necessarily lead to far-off p-values. In the light of intransparent guidance on which test to choose for this specific case, I can understand if one sticks to the t-test, but one should include a note on its limitations. [Note: Permutation tests (requiring access to instance-level performance scores), might be a valid alternative. The use of popular other methods like the Wilcoxon Signed Rank test would violate other assumptions; in this case the independence of same-model associated scores, which come from (bootstrapping the) same testing set samples.]
> >     - (b) When performing multiple tests on the same family of models, correction for multiple comparisons should be applied, or a suited multi-sample test. Otherwise, the resulting p-values do not represent the probability of the respective scores under the null hypothesis validly.
> >     - (c) A one-sided test is often considered suggestive of p-hacking and should usually only be used if the direction of comparison is sensible, was defined a priori, and mirrors the derived result message. In this context, this does not appear to be given.
> >     - (d) Significance should be indicated in the results tables using some symbol for it, or the details should be added to App. C.
> >
> > - Sec. 3.3: "For time-order based pretraining, the RnC:t baseline performs worse than ChronoCon". If not underlined statistically, this should be relaxed, saying "performs numerically [or slightly] worse than", or similar. I cannot tell from the local context if there was significance due to missing test indicators, see above.
> > - Sec. 2 (end): Here, or in the appendix, it could be valuable to state that the t-test was done "on the boostrapped MSE scores", since "on MSE" leaves a bit of space for misunderstanding, if I understood the testing procedure correctly.
> > - Typo: "for the later" in Sec 3.2.

---

> ### Author Response · Authors · 2026-01-31
> **Improvements on statistical analysis and description thereof**
>
> We thank the referee again for many good suggestions to improve the presentation and for a very useful reference.  Some remarks are in order.
>
> - **(1c) One-sided vs two-sided test**:
> We agree and follow the suggested approach by changing all one-sided tests to a two-sided version. The overall story is not changed by this. All p-values below correspond to the one-sided tests. We propose to change this in the final manuscript as well.
>
>
> - **(1a) Alternative to t-test** We agree with the referee that paired t-tests on the same test set can underestimate variance and should be interpreted with care. We will therefore add a short note in the manuscript explicitly stating this limitation and clarifying that the reported p-values should be interpreted as approximate evidence rather than exact probabilities.
> To verify robustness of the results, we additionally performed permutation tests based on instance-level MSE differences.
>     - *Single-stage baseline*   vs   *ChronoCon*
>          - cross sectional:  t-test p < 1e-4      ;   permutation test: p < 1e-4
>          - longitudinal:       t-test p < 1e-4      ;   permutation test:   p < 1e-4
>     -  *RnC*  vs   *OrdinalCon:Y*
>          - cross sectional:  t-test p=0.016          ;   permutation test: 0.013
>          - longitudinal:       t-test p=0.14 (n.s)   ;   permutation test:    0.15  (n.s)
>     - *RnC:t*   vs   *ChronoCon*
>          - cross sectional:  t-test p=0.016      ;   permutation test:    p = 0.012
>          - longitudinal:       t-test p=0.025      ;   permutation test:   p = 0.023
>     - *RnC:t*  vs   *ChronoCon*  at **4% of labels** (20 patients)
>         - cross sectional:  t-test p<1e-4      ;   permutation test:    p<1e-4
>         - longitudinal:       t-test p<1e-4       ;   permutation test:    p<1e-4
>
>    The resulting p-values were qualitatively unchanged and numerically very similar to the paired t-tests across all reported comparisons. Given this close agreement, and in the interest of keeping the presentation concise, we prefer to retain the t-test results in the manuscript while explicitly noting their limitations and mentioning permutation testing as a viable alternative.
>
> - **(1b)** We acknowledge that multiple comparisons can inflate type-I error. Our goal with the tests was not to establish formal confirmatory claims across a family of hypotheses, but to provide quantitative support for observed performance differences. We will clarify this in the manuscript and avoid wording that suggests strict hypothesis dismissal.
>
>
> - **(1d) add markers to table 2** We agree that statistical significance should be indicated more clearly. We will add appropriate significance markers to the result tables and/or provide the detailed p-values in Appendix C, subject to the final formatting constraints.
>
> - **(2)** We intend to soften the wording as suggested in the camera-ready version.
>
> - **(3)** We thank the referee for pointing this out, as this section indeed requires clarification.
> All hypothesis tests (t-test and permutation test) were performed on the paired, per-instance MSE differences on the fixed test set. Bootstrapping was used only for confidence intervals of ICC/RMSE and was never used for hypothesis testing.
>
>
> - **(4)** Thank you for spotting the typo – we will correct “for the later” in Sec. 3.2.

---

### Official Review · Reviewer_aFRd · 2026-01-09

**Confidence:** 3
**Preliminary Rating:** 4
**Final Rating:** 4

**Summary:**

The paper introduces ChronoCon, a self-supervised contrastive learning framework that uses visitation order in longitudinal RA imaging to learn disease-relevant representations without expert annotations. Under the assumption of monotonic progression in irreversible diseases, it replaces label-based ranking with temporal ordering and shows notable gains in low-label and few-shot settings.

**Strengths:**

1) The paper cleverly exploits an inexpensive, widely available signal(the visit ordering) to impose ordering constraints on learned features. which is practically useful for irreversible progression.
2) ChronoCon yields substantial gains in low-label settings, including convincing few-shot results and strong performance on longitudinal change, which is clinically meaningful and often overlooked. The results on both severity prediction and disease progression are convincing and practically important.
3) The loss formulation, two-stage training setup, and ablation studies are well presented. It includes insightful feature-space analyses supporting their main claims.

**Weaknesses:**

1) All experiments are conducted on a single-center rheumatoid arthritis dataset. It is unclear how well the method would generalize to data from other sources or imaging settings.
2) The paper mainly compares against variants of ranking-based losses. A comparison with standard self-supervised image methods would help better understand how much benefit comes specifically from using the visit order.
3) The approach assumes mostly monotonic disease progression. While this is reasonable for irreversible diseases, it limits the use of the approach for non-monotonic or treatment-driven changes.

**Detailed Comments:**

Please refer to the Weaknesses section.

**Justification Of Final Rating:**

The authors have addressed the primary concerns regarding statistical significance and benchmarking. The inclusion of the SimCLR and RnC:t baselines effectively isolates the benefit of the chronological contrastive loss, particularly in low-label settings where the method shows substantial gains. While the study currently utilises a single-center dataset, leaving room for future validation across broader populations and acquisition protocols, the results demonstrate the core methodological benefits. Similarly, while the monotonicity assumption naturally focuses the clinical application on irreversible conditions, it provides a powerful and practical inductive bias for those specific disease types. Future work exploring more varied disease trajectories would further expand the practical impact of this otherwise promising self-supervised approach.

**Justification Of The Preliminary Rating:**

The paper presents a clear and well-motivated approach that leverages chronological supervision to achieve strong label efficiency in longitudinal medical imaging. The use of visit ordering provides a practical and inexpensive inductive bias, particularly suitable for irreversible disease progression. However, as mentioned above, addressing these points would further strengthen the paper.

**Questions To Address In The Rebuttal:**

Please refer to the Weaknesses section.

---

> ### Author Response · Authors · 2026-01-24
> **Added SSL baseline, discussion of limitations, and monotonicity assumption**
>
> We thank the referee for the careful evaluation of our paper, for appreciating the clinical relevance and presentation, and for providing constructive detailed feedback.
>
> In the following we respond comment by comment:
>
> ----
>
> > All experiments are conducted on a single-center rheumatoid arthritis dataset. It is unclear how well the method would generalize to data from other sources or imaging settings.
>
> We acknowledge that the use of a single-center dataset is a limitation of our work. We believe that the principled benefits of the proposed methods are demonstrated by the current single-center, but relatively large expert annotated data set. Having said this, we plan to expand the evaluation to other data sets in future work.
>
> > The paper mainly compares against variants of ranking-based losses. A comparison with standard self-supervised image methods would help better understand how much benefit comes specifically from using the visit order.
>
> We agree and have added additional experimental results adding *SimCLR* [Chen et al., ICML 2020]* as an additional SSL baseline (see Figure 3; Table 2). Choosing SimCLR allowed us to keep the ResNet18 backbone used throughout this manuscript. Thus, the comparison is on equal footing. We also added statistical analysis of the difference between methods. In low-label setting our method outperforms all other methods including *SimCLR*. Moreover, we added the original RnC method [*Zha et al., NeurIPS 2023*] applied to *time* (called *RnC:t*) as an additional baseline/ablation study. This method also uses visitation order but is only applicable when there is a mostly linear relationship between time and disease progression. It performs better than SimCLR but worse than the proposed ChronoCon method. These additional results show that the benefit originates from utilizing visitation-order information and that, in this monotonic but non-linear disease progression setting, our method leads to improvements.
>
>
> > The approach assumes mostly monotonic disease progression. While this is reasonable for irreversible diseases, it limits the use of the approach for non-monotonic or treatment-driven changes.
>
> We acknowledge that the monotonic progression assumption is not fulfilled for many diseases. However, a substantial group of irreversible diseases  and other processes (e.g., material wear) exhibits monotonic progression, and thus it provides a useful inductive bias. Medical examples we have in mind include many neurodegenerative diseases, which are typically irreversible. Beyond disease progression, we hypothesize that infant development might also be an interesting test case for our method.
> While we acknowledge this limitation of our work, we believe that the approach can find application in many areas, especially in the medical imaging domain, where annotations are often scarce due to the expertise required for accurate labeling.
>
> ---
>
> We hope that our clarifications and additional experiments sufficiently address the reviewer’s concerns and further demonstrate the practical relevance and methodological soundness of ChronoCon.

---

### Official Review · Reviewer_Pewb · 2026-01-10

**Confidence:** 4
**Preliminary Rating:** 3

**Summary:**

This paper presents a chronological contrastive learning objective function to learn to extract progression-related latent representation and quantitative severity score without severity labels. This is achieved in a two-stage learning method, where the first stage learns a progression-aware latent representation, and the second stage learns to map the representation to an estimate of the ordinal score. The idea of the contrastive learning leverages an assumption of “monotonic progression”. The method is tested on RA radiographs

**Strengths:**

The idea of leveraging the order of longitudinal images to extract progression-aware latent representations is interesting, and the use of contrastive learning as presented seems an interesting way to achieve it.

The addition of Chrono loss to a MSE loss showed evident gain of performance, especially when the percentage of labels is low (less than 15%).

**Weaknesses:**

Results in Table 2 did not seem to show significant gains of performance by the chronological contrastive loss. Even in cases where it showed better performance, the gain seems to be at the order or smaller than the standard deviation.

The presentation of results in Fig 4 and Fig 5c are difficult to interpret, due to both the size of the figures and the way the information is presented.

**Detailed Comments:**

Some of the figures, such as Fig 4 and Fig5c, are very difficult to read and interpret.

**Justification Of The Preliminary Rating:**

The idea behind the chronological contrastive loss is interesting and has good application potential. The results related to labeling efficiency was promising. The unclear gain of performance in Table 2 however brings some question about the importance of this objective function, and the presentation of some of the figures hampers the assessment of the results.

**Questions To Address In The Rebuttal:**

Clarifications to the performance differences in Table 2 could play a major role in my final rating of the paper.

---

> ### Author Response · Authors · 2026-01-24
> **Clarify ChronoCon strengths, table 2 significance analysis and figure improvements**
>
> We would like to thank the reviewer for the careful evaluation of our manuscript and for highlighting the interesting nature of both the core idea and the proposed contrastive approach.
>
> In the following, we respond to each comment in detail:
>
> ---
>
> > Results in Table 2 did not seem to show significant gains of performance by the chronological contrastive loss. Even in cases where it showed better performance, the gain seems to be at the order or smaller than the standard deviation.
>
> We agree with the referee that the performance differences between the different loss variations were presented suboptimally. In the revised manuscript, we now emphasize that the main advantage of our method is label efficiency (Table 1). When all images come with expert annotations, this appears to be such a strong supervision signal to the model that the improvements due to pretraining are comparatively small.
> While these differences are numerically not very large, we now also performed a statistical test (one-sided paired t-test) on the mean squared errors (MSE).
>
> - ChronoCon+DAE vs. single-stage baseline: statistically significant improvements
> (p = 5e-6 longitudinally; p = 4e-5 cross-sectionally)
> - ChronoCon+DAE vs. best baseline (RnC) in the full-label setting: no statistically significant difference
> - OrdinalCon:Y (label ordering) vs. best baseline (RnC): statistically significant for cross-sectional evaluation only
>  (p = 0.07 longitudinally; p = 0.008 cross-sectionally)
>
> While we agree with the referee that the numerical improvements are not large, we would like to emphasize that they are statistically significant compared to the single-stage baseline.
> However, in the low-label setting, our method does lead to large improvements over all other methods. In the revised manuscript, we clarified the performance differences in Table 2 by improving the text, restructuring the table, and adding two additional baselines (*SimCLR* [Chen et al., ICML 2020] and *RnC:t* [Zha et al., NeurIPS 2023]) . We hope this clearer presentation and the additional analysis make the performance differences easier to interpret.
>
> > The presentation of results in Fig 4 and Fig 5c are difficult to interpret, due to both the size of the figures and the way the information is presented.
>
> We now increased the size (+ legend size)  of figure 4 and 5c, extended the caption text and improved the corresponding section in the main text.
>
> ---
>
>
> We hope that the clarifications, additional baselines, statistical analysis, and improved presentation address the concerns raised in the review. We would be grateful if the reviewer could reconsider the rating in light of the revised manuscript.

---

> > ### Comment · Reviewer_Pewb · 2026-02-02
> >
> > I'd like to thank the authors for providing the clarifications especially detailed statistical significance test. I'd improve my rating to "Weakly accept".

---

### Author Rebuttal · Authors · 2026-01-24

**Rebuttal:**

We thank the reviewers for the careful consideration of the manuscript and their comments that helped us to improve it. In the following we briefly summarize responses to the main points raised by the reviewers, and provide detailed explanations in the responses to individual reviewers.

**Benefit of the method:** the primary benefit of the method is label efficiency, i.e., exploiting large scale data without labels to improve the performance of models trained on few examples with labels. We show that the embedding space learned in an unsupervised manner enables a model fine tuned on few labeled examples to achieve relevant labeling accuracy, and consistently outperforming models trained only on the supervised data.

While in fully labeled data the proposed approach performs comparably to state-of-the-art methods, it outperforms them in case of reduced label availability.

**Single center data**: we appreciate the limitation of a single center data set, even-though it has relatively large size. We have clarified the manuscript accordingly. However, we believe that the evaluation results are sufficient to assess the benefits of the proposed methodological approach.

**Code release**: we have released the code on  https://github.com/cirmuw/ChronoCon

**Additional experiments and baselines**: As suggested by referee aFRd and 9yrsM  we added two additional setups to the updated manuscript: A self-supervised baseline which does not have access to temporal information (*SimCLR*), and Rank-N-Contrastive learning on time (*RnC:t*) to check how much non-linear progression plays a role in learning a good representation.

Detailed responses including a detailed discussion of the additional experiments are provided in the individual reviewer response fields.
Note that we attached a **revised manuscript**, as well as a **latexdiff_rebuttal_v2_vs_v1.pdf**  where all changes are marked, for the referee's convenience.

**Supporting Material:**

/attachment/8e681cebe58a14b0ae85d54ca4fe18466c6ff200.zip

---

### Meta-Review · Area_Chair_SKdN · 2026-02-03

**Recommendation:** Accept (Poster)
**Confidence:** 4

**Metareview:**

In the rebuttal stage, the authors have well addressed the primary concerns regarding statistical significance and benchmarking. All reviewers recognised its contribution and clarity, and suggested acceptance. The AC agrees with them.

---

### Decision · Program_Chairs · 2026-02-13

Accept (Poster)